# Osteoblast-Osteoclast Communication and Bone Homeostasis

**DOI:** 10.3390/cells9092073

**Published:** 2020-09-10

**Authors:** Jung-Min Kim, Chujiao Lin, Zheni Stavre, Matthew B. Greenblatt, Jae-Hyuck Shim

**Affiliations:** 1Division of Rheumatology, Department of Medicine, University of Massachusetts Medical School, Worcester, MA 01605, USA; jungmin.kim@umassmed.edu (J.-M.K.); chujiao.lin@umassmed.edu (C.L.); zheni.stavre@umassmed.edu (Z.S.); 2Department of Pathology and Laboratory Medicine, Weill Cornell Medical College, New York, NY 10065, USA; mag3003@med.cornell.edu; 3Li Weibo Institute for Rare Diseases Research, University of Massachusetts Medical School, Worcester, MA 01605, USA

**Keywords:** bone, osteoblast, osteoclast, bone remodeling

## Abstract

Bone remodeling is tightly regulated by a cross-talk between bone-forming osteoblasts and bone-resorbing osteoclasts. Osteoblasts and osteoclasts communicate with each other to regulate cellular behavior, survival and differentiation through direct cell-to-cell contact or through secretory proteins. A direct interaction between osteoblasts and osteoclasts allows bidirectional transduction of activation signals through EFNB2-EPHB4, FASL-FAS or SEMA3A-NRP1, regulating differentiation and survival of osteoblasts or osteoclasts. Alternatively, osteoblasts produce a range of different secretory molecules, including M-CSF, RANKL/OPG, WNT5A, and WNT16, that promote or suppress osteoclast differentiation and development. Osteoclasts also influence osteoblast formation and differentiation through secretion of soluble factors, including S1P, SEMA4D, CTHRC1 and C3. Here we review the current knowledge regarding membrane bound- and soluble factors governing cross-talk between osteoblasts and osteoclasts.

## 1. Introduction

Bone is a dynamic tissue that remodels continuously throughout life, providing mechanical support for stature and locomotion and protecting vital organs such as bone marrow and the brain [1]. Bone also functions as a reservoir for calcium and phosphate. Continuous remodeling is required to preserve both of these critical functions by preventing accumulation of bone damage and maintaining both the mechanical strength of bone and calcium homeostasis [2,3].

Bone remodeling is a process in which old or damaged bone is removed by osteoclasts and replaced with new bone formed by osteoblasts. Osteoclasts, bone-resorbing cells, originate from hematopoietic stem cells (HSCs) [4,5,6,7,8] and degrade bone via secretion of acid and proteolytic enzymes, such as cathepsin K (CTSK), that dissolve collagen and other matrix proteins during bone resorption [9,10]. Osteoblasts, bone-forming cells, arise from the commitment of mesenchymal precursors to osteoprogenitor lineages through the sequential action of transcriptional factors and terminally differentiate into osteocytes [11,12,13,14]. Osteoblasts produce extracellular proteins, including osteocalcin, alkaline phosphatase and type I collagen, the latter of which makes up over 90% of bone matrix protein. The extracellular matrix is first secreted as unmineralized osteoid and subsequently mineralized through the accumulation of calcium phosphate in the form of hydroxyapatite [15]. The sequential strategies of osteoclastogenesis and osteoblastogenesis are shown in Figure 1.

Bone remodeling is traditionally considered to be composed of four sequential phases [16]: the activation phase when osteoclast progenitors are recruited to damaged bone surface; resorption phase when mature osteoclasts resorb damaged bone; reversal phase when osteoclasts die and osteoblast progenitors are recruited; formation phase when mature osteoblasts produce new bone matrix (osteoid) and this matrix is mineralized [17,18]. Almost all new bone formation is observed in areas with previous resorption and in distinct anatomical structures called basic multicellular units (BMUs) [19]. The balance between osteoblast-mediated bone formation and osteoclast-mediated bone resorption is tightly regulated without a major alteration in a net bone mass or mechanical strength under homeostatic conditions [2]. However, dysregulation of this balance results in abnormal bone remodeling, resulting in both postmenopausal and secondary forms of osteoporosis, such as diabetes-associated and glucocorticoid-induced osteoporosis [20,21,22]. In addition to improving understanding of the bone resorption and formation phases, a more detailed study on the reversal phase might be necessary as reversal step dysfunction is associated with pathologic bone loss [23,24].

In this review, we illustrate the key mediators that control the cross-talk between osteoblasts and osteoclasts through cell–cell contact or secretory factors (Figure 2).

The factors which control osteoblast–osteoclast communication will be reviewed in cellular levels, mainly in genetic mouse models.

## 2. Membrane-Bound Mediators of Cell-to-Cell Communication

### 2.1. EFNB2 (Ephrin B2)-EPHB4

During bone remodeling, osteoblasts and osteoclasts communicate through cell-to-cell direct contact [25]. This interaction can be mediated by Ephrin signaling, which is critical for the bidirectional communication between osteoclasts and osteoblasts [26,27,28]. Cell-surface molecules Ephrin B (B1∼B3) bind to their cognate tyrosine kinase receptors EPHB (B1∼B6). The Ephrin B family consists of transmembrane proteins with cytoplasmic domains and its interaction with EPHB-expressing cells mediates bidirectional signal transduction [29]. Ephrin B2 (EFNB2), expressed on the cell surface of osteoclasts, binds to osteoblast surface molecule EPHB4. Reverse signaling (osteoblast to osteoclast) is initiated by EPHB4-mediated activation of EFNB2 and suppresses osteoclast differentiation by blocking the osteoclastogenic C-FOS/NFATC1 cascade. In forward signaling (osteoclast to osteoblast), EFNB2-mediated activation of EPHB4 promotes osteoblast differentiation and suppresses apoptosis [30]. Similarly, overexpression of EPHB4 in osteoblasts increases bone mass in a transgenic mouse model [26].

### 2.2. FAS Ligand (FASL)-FAS

FAS (also called as APO-1 or CD95) is a death domain–containing member of the tumor necrosis factor receptor (TNFR) superfamily. FAS ligand (FASL or CD95L) plays a central role in the physiological regulation of apoptosis of FAS-expressing cells, which is associated with various disease processes, such as tumorigenesis and several auto-immune diseases [31,32]. Deficiency of estrogen is closely linked to postmenopausal osteoporosis, and estrogen has been known to induce osteoclast apoptosis [33,34]. In line with the effects of estrogen on osteoclast apoptosis, further study demonstrated that FAS-FASL signaling is involved in estrogen-induced osteoclast apoptosis [35]. Upregulated expression of FASL in osteoblasts by estrogen results in apoptosis of pre-osteoclasts, suggests that a paracrine signal emanating from osteoblasts plays an important role in the protective effects of estrogen in bone. Additional study has demonstrated that conditional knockout of FASL in osteoblasts increases the number and activity of osteoclasts, resulting in reduced bone mass [36]. Similarly, osteoblasts from ovariectomized (estrogen-deficient) mice exhibit a decrease in FASL expression, resulting in reduced osteoclast apoptosis and increased bone resorption.

### 2.3. Semaphorin 3A (SEMA3A)-NRP1

Semaphorins were originally identified as axon guidance factors involved in the development of neuronal system, but they also play a role in various physiological processes including osteoblast-osteoclast interactions [37,38,39]. Semaphorin 3A (SEMA3A) is the first semaphorin family member identified in vertebrates and ubiquitously expressed in many types of tissue, including brain and bone. While SEMA3A participates in the development of major structures of the nervous system [40], many studies suggest that SEMA3A also plays a key role in both bone modeling and remodeling [40,41,42]. In particular, sensory neuron-derived SEMA3A is necessary for normal bone formation in vivo [42]. Additionally, SEMA3A, produced by osteoblast lineage cells, functions as a potent osteoprotective factor by synchronously inhibiting bone resorption and promoting bone formation [38]. High levels of SEMA3A were detected in the conditioned media of osteoprotegerin (OPG, anti-osteoclastogenic factor)-deficient osteoblasts, and its binding to neuropilin-1 (NRP1) inhibits RANKL-induced osteoclast differentiation, and promotes osteoblast differentiation through the WNT/β-catenin pathway.

## 3. Soluble Factors Released from Osteoblasts

### 3.1. Macrophage Colony-Stimulating Factor (M-CSF)

M-CSF (also called as CSF1) is a hematopoietic growth factor that allows for survival, proliferation, differentiation, and mobility of mononuclear phagocyte lineages, including osteoclasts [43,44]. M-CSF, secreted from osteoblasts and bone marrow stromal cells, binds to its cognate receptor C-FMS on the surface of osteoclasts and monocytes/macrophages [45]. Osteopetrotic (op/op) mice where a thymidine insertion in the *Csf1* gene resulted in M-CSF deficiency show decreased numbers of macrophages and osteoclasts at a young age. However, these phenotypes disappear during aging. Injection of recombinant M-CSF or production of soluble M-CSF in osteoblasts increases osteoclast numbers and rescues osteopetrotic phenotypes in op/op mice, demonstrating that M-CSF is crucial for osteoclast formation at least in young mice, but does not exclude the existence of M-CSF-independent compensatory mechanisms.

### 3.2. Receptor Activator of NF-κB (Nuclear Factor-Kappa B) Ligand (RANKL)

RANKL is also called osteoclast differentiation factor (ODF), TNF ligand superfamily member 11 (TNFSF11), TNF-related activation-induced cytokine (TRANCE), and OPG ligand (OPGL) [46,47]. RANKL is highly expressed in osteoblasts, osteocytes, activated T lymphocytes, and lymph nodes [48,49,50]. RANKL binds to its cognate receptor, receptor activator of NF-κB (RANK) on the surface of osteoclasts and osteoclast precursors, leading to osteoclast differentiation, fusion, and activation [6,51]. Mice deficient in *Tnfrsf11a* (RANK) or *Tnfsf11* (RANKL) are phenocopies of one another, indicating the essential role of this RANKL/RANK signaling axis in bone remodeling [48,51]. Deletion of RANKL in mice results in severe osteopetrosis due to absence of osteoclasts, whereas overexpression of soluble RANKL leads to severe osteoporosis [48,52]. Accordingly, blocking RANKL signaling has been proposed as a promising therapeutic target for osteoporotic bone loss and related skeletal disorders.

### 3.3. Osteoprotegerin (OPG)

OPG is also known as osteoclastogenesis inhibitory factor (OCIF) and TNF receptor superfamily member 11B (TNFRSF11B) [47,53,54]. OPG was identified as a secreted glycoprotein synthesized by many types of cells, including osteoblasts, lung- or liver-residing cells, and B lymphocytes in the bone marrow [53,54,55]. Overexpression of OPG results in profound osteopetrosis due to inhibition of osteoclast formation, whereas *Tnfrsf11b* (OPG)—deficient mice exhibit rapid postnatal bone loss and severe bone porosity due to an increased osteoclast development [53,56]. OPG is considered to function as a decoy receptor binding to RANKL, negatively regulating osteoclast differentiation and activation by blocking the RANKL-RANK interaction [6,57].

### 3.4. WNT5A

The WNT pathway is crucial for the maintenance of bone homeostasis by regulating osteoblastogenesis and osteoclastogenesis through both β-catenin-dependent (canonical) and -independent (noncanonical) pathways [58]. A noncanonical WNT ligand, WNT5A, is highly expressed in osteoblast-lineage cells and binds to its cognate receptor, receptor tyrosine kinase-like orphan receptor 2 (ROR2), on the surface of osteoclasts [59]. Heterozygous deletion of *Wnt5a* or *Ror2* in mice resulted in impaired development of bone marrow-derived monocytes (BMM) to mature osteoclasts. Corresponding defects in osteoclastogenesis were also observed in mice with osteoblast-specific deletion of *Wnt5a* or osteoclast-specific deletion of *Ror2*. WNT5A enhances RANKL-induced osteoclastogenesis by upregulating RANK expression in osteoclasts via activation of the Jun–N-terminal kinase (JNK) MAPK pathway.

### 3.5. WNT16

The *WNT16* locus is closely associated with bone mineral density (BMD), cortical bone thickness, and fracture risk in humans [60,61,62]. WNT16 is highly expressed in osteoblast-residing cortical bone, but little to no expression is detected in osteoclasts [63]. Global deletion of *Wnt16* results in a specific decrease in cortical bone mass and an increase in cortical porosity, along with spontaneous fractures where there is no alteration in trabecular bone. WNT16 suppresses osteoclastogenesis in both a direct and indirect manner. In addition to direct inhibition of osteoclastogenesis via the noncanonical JNK MAPK pathway, WNT16-induced phosphorylation of JUN upregulates expression of OPG in osteoblasts, providing a direct mechanism to suppress osteoclastogenesis. Osteoblast-specific deletion of *Wnt16* in mice phenocopies mice with its global deletion, suggesting osteoblasts are a primary source of WNT16, with an impact on cortical bone and skeletal integrity. Table 1 provides a list of osteoblast-derived factors that regulate osteoclasts.

## 4. Soluble Factors Released from Osteoclasts

### 4.1. Sphingosine 1 Phosphate (S1P)

Sphingosine kinase (SPHK), expressed in osteoclasts, phosphorylates sphingosine to generate sphingosine 1 phosphate (S1P) promotes osteoblastogenesis [64,65]. S1P production is enhanced in osteoclast precursors upon RANKL stimulation and S1P binds to S1P receptor on the surface of osteoblasts, resulting in increased osteoblast migration and survival. Subsequently, S1P-activated osteoblasts upregulate RANKL expression, augmenting osteoclastogenesis [66]. Further study has demonstrated that osteoclast-specific deletion of *Ctsk* (cathepsin K), the lysosomal cysteine protease that degrades type 1 collagen and matrix proteins during bone resorption showed enhanced bone formation by increasing S1P production from osteoclasts [65,67].

### 4.2. Semaphorin 4D (SEMA4D)

SEMA4D is an axon guidance molecule that is also expressed in osteoclasts. Osteoclast-derived SEMA4D binds to Plexin-B1 (PLXNB1) on the surface of osteoblasts, suppressing osteoblast differentiation [39]. Mice lacking *Sema4d* show high bone mass with increased bone forming activity and enhanced bone strength. *Plxnb1*-deficient mice also show similar bone phenotypes to those seen in *Sema4d*-deficient mice, suggesting that SEMA4D and PLXNB1 are in the same pathway. Mechanistically, binding of SEMA4D to PLXNB1 activates small GTPase RHOA and subsequently suppresses osteoblast differentiation through attenuation of insulin-like growth factor-1 (IGF-1) signaling. Administration of an anti-SEMA4D antibody prevents bone loss in a mouse model of postmenopausal osteoporosis and promotes bone formation without affecting osteoclast-mediated bone resorption, suggesting that SEMA4D is a potential therapeutic target for osteoporosis or other low bone mass disorders [68].

### 4.3. Collagen Triple Helix Repeat Containing 1 (CTHRC1)

CTHRC1 is a soluble protein released from mature osteoclasts that targets stromal cells to induce osteoblast differentiation [69]. *Cthrc1* expression is upregulated when mature osteoclasts contact hydroxyapatite and calcium. Although the CTHRC1 receptor in the osteoblast has not yet been identified, recombinant CTHRC1 can induce recruitment of stromal cells and osteoblastic differentiation, promoting bone formation. In a line with this, osteoclast-specific deletion of *Cthrc1* results in low bone mass with reduced bone formation.

### 4.4. Complement Component C (C3)

Osteoclast-derived C3 is cleaved to C3a that induces osteoblastogenesis [70]. Expression of C3 is upregulated during osteoclastogenesis and a C3a receptor (C3AR) antagonist blunts the osteogenic activity of osteoclast-derived conditioned medium, while a C3AR agonist promotes osteoblast differentiation. C3 expression in bone is markedly upregulated in the setting of ovariectomy (OVX)-induced osteoporosis, or when administered with RANKL. In these settings, a C3AR antagonist attenuates bone formation activity, accelerating bone loss.

### 4.5. Other Factors

WNT10B activates the canonical WNT signaling pathway to promote osteoblast differentiation [71]. Osteocalcin promoter-driven expression of *Wnt10b* in osteoblasts increases bone mineral density and trabecular bone volume and number, whereas *Wnt10b*-deficient mice show a decrease in trabecular bone mass and serum levels of osteocalcin [71,72]. Additionally, osteoclast-derived WNT10B increases mineralization of human mesenchymal stromal cells, which suppress the presence of recombinant DKK1 (dickkopf WNT signaling pathway inhibitor 1), a WNT antagonist [65]. BMP6 in osteoclast-conditioned media induces migration and mineralization of human mesenchymal stromal cells, suggesting it as a coupling factor between osteoclasts and osteoblasts [65]. Finally, vesicular RANK has been considered as a coupling factor [73]. Vesicular RANK secreted from osteoclasts binds to osteoblastic RANKL and promotes osteoblast differentiation and bone formation by RANKL reverse signaling. RANKL is initially produced as transmembrane protein and cleaved by protease to yield a soluble form (sRANKL) and this sRANKL is required for osteoclast formation in cancellous bone as the skeleton matures [74]. Mostly, the membrane-bound form of RANKL is sufficient for osteoclast formation in developing skeleton. Accordingly, vesicular RANK might have different acting mechanism during osteoblast differentiation and bone development. Table 2 provides a list of osteoclasts-derived factors to influence on osteoblasts behavior.

## 5. Matrix-Derived Coupling Factors by Bone Resorption

### 5.1. Transforming Growth Factor β1 (TGF-β1)

TGF-β1, one of the most abundant proteins in the bone matrix, contributes to bone remodeling through regulation of both osteoblasts and osteoclasts. In the bone matrix, TGF-β1 is non-covalently bound to the latency-associated protein (LAP), keeping it in latent state by masking the receptor-binding domains of the TGF-β1 [75,76]. Accordingly, TGF-β1 stays at an inactive state in bone matrix and is released from bone matrix in response to osteoclastic bone resorption [77]. Active TGF-β1 recruits bone mesenchymal lineage cells to resorptive surfaces and differentiates them into bone-forming osteoblasts.

### 5.2. Insulin-Like Growth Factor Type 1 (IGF-1)

IGF-1 is another growth factor deposited in the bone matrix, bound to insulin-like growth factor-binding protein (IGFBP) [78,79]. IGF-1 released from bone matrix is activated by acid pH during osteoclastic bone resorption [80]. Bone matrix-derived IGF-1 was found to promote osteogenesis through activation of mammalian target of rapamycin (mTOR) in osteoblast lineage cells [81]. Of note, IGF-1 concentration in the bone matrix is lower in aged rats than in young rats, suggesting a correspondence between bone volume and IGF-1 levels.

## 6. Impact of Therapeutic Agents on Osteoblast-Osteoclast Interactions

Despite there being many factors identified that regulate osteoblast-osteoclast interactions, currently very few of these pathways have resulted in approved therapeutics for the treatment of skeletal disorders. However, osteoblast–osteoclast interactions have played a powerful role in shaping the action of all currently approved drugs acting on the skeleton, including often imposing limitations on the activities of these agents. Most anti-resorptive agents inhibiting osteoclast formation and activity simultaneously suppress bone formation, whereas the activity of anabolic agents inducing bone formation are similarly tempered by simultaneously increasing bone resorption.

Denosumab is a humanized monoclonal antibody that binds RANKL and acts as an OPG-like RANK decoy receptor, disrupting the RANKL-RANK interaction and osteoclast differentiation, function, and survival, leading to a decrease in bone resorption. In patients, biannual administration of Denosumab increases bone density, and reduces the risk of vertebral, nonvertebral, and hip fractures [82]. However, as expected, inhibiting osteoclast formation by blocking RANKL-RANK signaling pathway results in reduction of osteoclast-derived coupling factors that induce osteoblast-mediated bone formation. Denosumab leads to an immediate and sustained decrease in markers of bone resorption such as urinary N-telopeptide (NTX) along with a delayed reduction in markers of bone formation, such as serum bone specific alkaline phosphatase (BSAP) [83,84].

Odanacatib, a selective cathepsin K (CTSK) inhibitor, suppresses osteoclast-mediated bone resorption [85,86]. CTSK is a cysteine proteinase, abundantly expressed in activated osteoclasts and secreted to extracellular space that cleaves the N-telopeptide of collagen [87]. *CTSK* mutation in humans causes pycnodysostosis, a rare skeletal dysplasia, characterized by short stature, an increase in the bone density of long bones, acroosteolysis of distal phalangers and skull deformities [88]. Despite the high bone mass phenotype, patients with pycnodysostosis suffer from bone fragility with a high incidence of pathological fractures [89]. Odanacatib has received a lot of attention as a potential osteoporosis drug since it inhibits osteoclast activity and function, rather than osteoclast generation, resulting in prevention of bone resorption with more modest overall effects on the ability of osteoblasts to form bone than most other anti-resorptive agents. It still remains unclear whether these effects reflect functional blockade of osteoclast-driven coupling processes or whether CTSK inhibition may have direct effects on CTSK-expressing periosteal mesenchymal progenitors [90,91]. Although odanacatib was highly effective in reducing fractures of postmenopausal women in clinical phase-3, its further development was discontinued due to concerns regarding risk of cerebrovascular events [92].

Parathyroid hormone (PTH), a bone anabolic agent, is normally secreted by the parathyroid glands and contributes to the maintenance of calcium homeostasis through its direct actions on bone cells. Intermittent PTH administration increases bone mass by recruiting mesenchymal stromal cells into osteoblast lineage and promoting the differentiation of osteoprogenitors into mature osteoblast [93,94]. Also, PTH downregulates sclerostin (SOST) expression from osteocytes, leading to a favorable environment for osteoblast differentiation and function through enhanced WNT signaling pathway [95,96]. However, continuous treatment results in an increase in net bone resorption due to increases in RANKL and decreases in the RANKL decoy receptor, OPG, in osteoblasts and osteocytes [97]. As seen in each of these examples, one of the most fundamental and enduring challenges in developing increasingly effective skeletal therapeutics is overcoming the osteoblast/osteoclast interaction effects that often blunt the activity of both anti-resorptive and anabolic drugs.

## 7. Conclusions and Future Perspectives

In this review, we provide the current knowledge of osteoblast-osteoclast communication in preserving bone homeostasis. Although multiple cells including osteocytes, chondrocytes, skeletal stem cells, endothelial cells, and immune cells play a role in controlling bone homeostasis, eventually they drive bone remodeling through the regulation of osteoblasts or osteoclasts. Accordingly, a broader understanding of the cell types regulating bone remodeling extending beyond osteoblast–osteoclast interactions will bring more insights and greater opportunities to develop therapeutics for skeletal disorders, including osteoporotic bone loss.

Traditionally, bone remodeling has been suggested as a sequential stepwise process, with initial osteoclast-mediated bone resorption followed by osteoblast-mediated bone formation [98]. However, osteoclast activity is not always restricted to specialized remodeling sites, and bone resorption is not always followed by osteoblast repopulation in the setting of pathologic skeletal disorders, such as inflammatory bone loss. Moreover, osteoclast-derived coupling can occur independent of bone resorption activity. Therefore, the spatial and temporal regulation of the bone remodeling process requires further elucidation, including further identification of the communication factors regulating osteoclast/osteoblast communication in physiologic and pathologic contexts.

## Figures and Tables

**Figure 1 cells-09-02073-f001:**
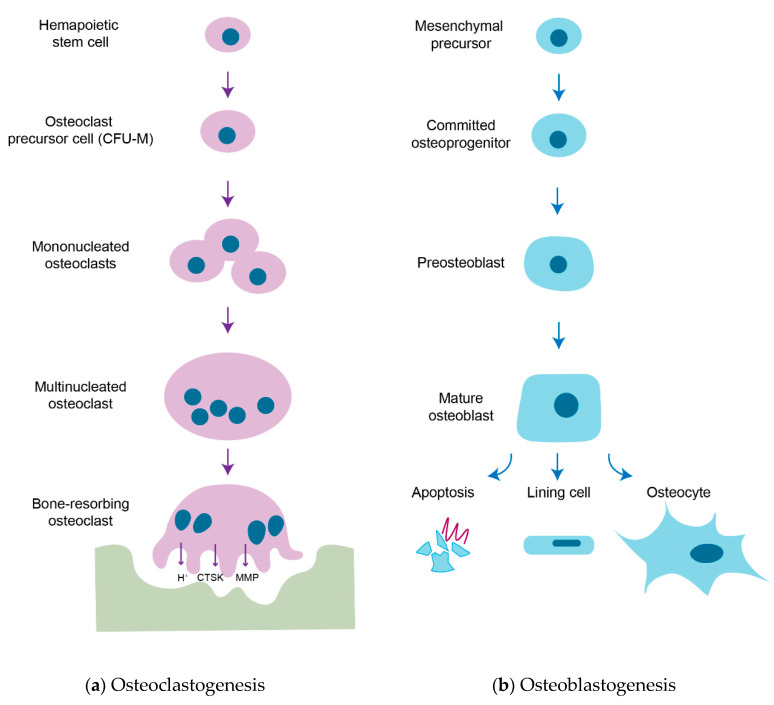
Strategies of osteoclastogenesis and osteoblastogenesis. (**a**) Osteoclastogenesis. Osteoclasts are tissue-specific macrophages derived from hematopoietic stem cells. In the presence of M-CSF, hematopoietic stem cells are committed to macrophage colony-forming units (CFU-M), the common precursor cells of macrophages and osteoclasts. When activated by the RANKL-RANK signal, CFU-M is further differentiated into mononucleated osteoclasts and subsequently fuse to become multinucleated osteoclasts. Multinucleated osteoclasts are fully matured upon a cognate interaction with osteoblasts and resorb bone matrix by secreting acids (H^+^), proteases (e.g., CTSK) and matrix metalloproteinases (MMPs) when they have a tight junction between the bone surface and basal membrane of osteoclasts to form a sealed compartment and then osteoclasts. (**b**) Osteoblastogenesis. Osteoblasts are derived from multipotent mesenchymal precursors and they are committed to osteoprogenitors and further differentiated into osteoblastic lineage through the expression of transcription factors RUNX2 and Osterix. They are continued to differentiation into matrix-producing mature osteoblasts and these cells have different fates: apoptosis, bone lining cells or osteocytes. A subpopulation of mature osteoblasts is surrounded by unmineralized osteoid and further differentiated into osteocytes, terminally differentiated bone cells in mineralized bone.

**Figure 2 cells-09-02073-f002:**
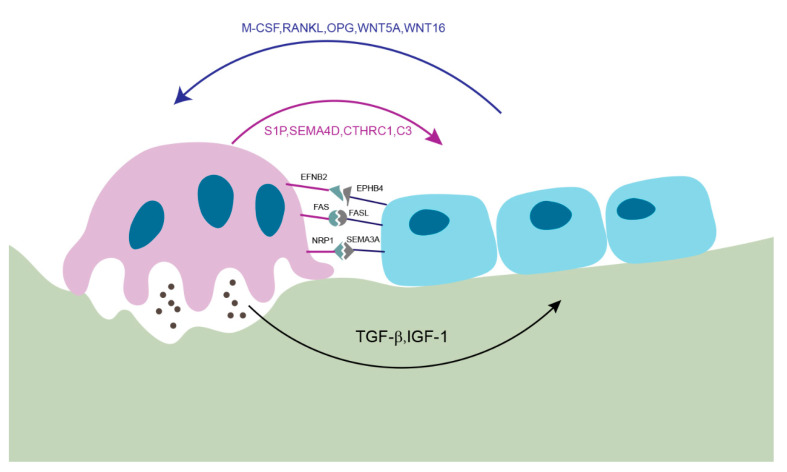
Key mediators of osteoblast-osteoclast interaction. Osteoblast-osteoclast communications are essential for fine-tuning of bone remodeling during bone homeostasis. (1) Osteoblasts and osteoclasts have direct contacts through the interactions between EFNB2-EPHB4, FAS-FASL and NRP1-SEMA3A to regulate cell proliferation, differentiation, and survival. (2) Osteoclast-mediated bone resorption releases TGF-β and IGF-1 from bone matrix to induce osteoblast-mediated bone formation. (3) Osteoblasts secrete M-CSF, RANKL, WNT5A that promote osteoclast formation and development and OPG and WNT16 that inhibit osteoclast activity. Conversely, osteoclasts secrete S1P, CTHRC1 and C3 that promote osteoblast differentiation and SEMA4D that suppresses osteoblasts differentiation.

**Table 1 cells-09-02073-t001:** Summary of the effect of osteoblast-derived factors on osteoclast behavior.

Osteoblast-Derived Factor	Mode of Action	Influences on Osteoclasts	References
EFNB2	Membrane-bound	Inhibits osteoclastogenesis	[26]
FASL	Membrane-bound	Induces osteoclast apoptosis	[36]
SEMA3A	Membrane-bound	Inhibits RANKL-induced osteoclastogenesis	[38]
M-CSF	Secreted	Promotes proliferation and survival of osteoclast precursor	[45]
RANKL	Membrane-bound and secreted	Promotes osteoclast differentiation and activation	[6,51]
OPG	Secreted	Inhibits osteoclastogenesis	[6,57]
WNT5A	Secreted	Promotes osteoclastogenesis	[58]
WNT16	Secreted	Inhibits osteoclastogenesis	[63]

**Table 2 cells-09-02073-t002:** Summary of effects of osteoclast-derived factors on osteoblasts.

Osteoclast-Derived Factor	Mode of Action	Influence on Osteoblasts	References
EPHB4	Membrane-bound	Promotes osteoblastogenesis and suppresses osteoblast apoptosis	[26,30]
S1P	Secreted	Promotes osteoblast migration and survival	[64,65]
SEMA4D	Secreted	Suppresses osteoblastogenesis	[39]
CTHRC1	Secreted	Recruits stromal cells and induces osteoblastogenesis	[69]
C3	Secreted	Promotes osteoblastogenesis	[70]
WNT10B	Secreted	Promotes osteoblastogenesis	[71,72]
Vesicular RANK	Secreted	Promotes osteoblastogenesis	[73]

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
