# Peer review of "Osteoblast-Osteoclast Communication and Bone Homeostasis"

_cells, 2020, doi:10.3390/cells9092073_

Round 1
Reviewer 1 Report
The review by Kim and co-authors gives a nice general and rather broad overview of various cellular interaction pathways between bone forming osteoblasts and bone resorbing osteoclasts during bone remodelling and homeostasis. The manuscript generally is very well written with intuitive figures. All major signalling pathways are covered, without going into much detail and depth of individual pathways. Besides various molecular pathways at play during bone remodelling also recent therapeutic developments are covered to treat osteoporosis and prevent bone fractures. However, the review also highlights the limitations of such approaches and stating the need for new development.
The manuscript would benefit from a thorough revision as it contains some incorrect grammar (e.g. Lines 86/87: Improve syntax and grammar).
Author Response
The review by Kim and co-authors gives a nice general and rather broad overview of various cellular interaction pathways between bone forming osteoblasts and bone resorbing osteoclasts during bone remodelling and homeostasis. The manuscript generally is very well written with intuitive figures. All major signalling pathways are covered, without going into much detail and depth of individual pathways. Besides various molecular pathways at play during bone remodelling also recent therapeutic developments are covered to treat osteoporosis and prevent bone fractures. However, the review also highlights the limitations of such approaches and stating the need for new development.
1) The manuscript would benefit from a thorough revision as it contains some incorrect grammar (e.g. Lines 86/87: Improve syntax and grammar).
- Thank you for the reviewer’s comment. We edited that sentence in the revised version (line 85-86).
Reviewer 2 Report
This review is very well written about the regulation of bone homeostasis by the communication of osteoblasts and osteoclasts.
Author Response
This review is very well written about the regulation of bone homeostasis by the communication of osteoblasts and osteoclasts.
- No reviewers' comments
Reviewer 3 Report
The review about Osteoblast-osteoclast communication and bone homeostasis is a clear overview of the communication during remodeling between the bone-related cells. However, some point needs to be clarified:
- The introduction starts at line 33 with osteoblasts followed by a part (line 40-64) about osteoclast and then again (line64-67). I think it will be easier to follow when started with osteoclast part and then both osteoblast parts.
- Line 70: resulting in postmenopausal and secondary forms of osteoporosis. Please explain secondary osteoporosis.
- Line 95: Ephrin E2 expressed in osteoclasts, binds to EPHB4 on the osteoblast surface. How can a protein in the cell connect with EPHB4 when it is not on the membrane? Or is EFNB2 also at osteoclast surface.
- Line 108: In line with : not in a
- Line 137: needs to be changed in osteoclasts at a young age.
- Line 138: please explain what you mean with is reversed over aging.
Author Response
The review about Osteoblast-osteoclast communication and bone homeostasis is a clear overview of the communication during remodeling between the bone-related cells. However, some point needs to be clarified:
1) The introduction starts at line 33 with osteoblasts followed by a part (line 40-64) about osteoclast and then again (line 64-67). I think it will be easier to follow when started with osteoclast part and then both osteoblast parts.
- Thank for the reviewer’s comment. As suggested, we switched the order of osteoblast parts with osteoclast parts in line 34-42 and Figure 1 and its legend.
2) Line 70: resulting in postmenopausal and secondary forms of osteoporosis. Please explain secondary osteoporosis.
- Thank you for pointing this out. As suggested, we added the explanation and examples of the secondary forms of osteoporosis in the revised manuscript (line 69-70, reference 22).
3) Line 95: Ephrin E2 expressed in osteoclasts, binds to EPHB4 on the osteoblast surface. How can a protein in the cell connect with EPHB4 when it is not on the membrane? Or is EFNB2 also at osteoclast surface.
- Yes, Ephrin B2 is a membrane-bound protein of osteoclast. We clarified this in the revised manuscript (line 94-95).
4) Line 108: In line with : not in a
- Thank you for the correction (line 107).
5) Line 137: needs to be changed in osteoclasts at a young age.
- Thank you for the correction (line 136).
6) Line 138: please explain what you mean with is reversed over aging.
- We clarified this sentence in the revised manuscript (line 136).